# Assessment of COVID-19 Molecular Testing Capacity in Jordan: A Cross-Sectional Study at the Country Level

**DOI:** 10.3390/diagnostics12040909

**Published:** 2022-04-06

**Authors:** Bara’a Qaqish, Malik Sallam, Maysa Al-Khateeb, Erik Reisdorf, Azmi Mahafzah

**Affiliations:** 1Abt Associates, United States Agency for International Development (USAID) Funded Local Health System Sustainability Project (LHSS), Amman 11822, Jordan; baraa_qaqish@abtassoc.com; 2Department of Pathology, Microbiology and Forensic Medicine, School of Medicine, The University of Jordan, Amman 11942, Jordan; 3Department of Clinical Laboratories and Forensic Medicine, Jordan University Hospital, Amman 11942, Jordan; 4Department of Translational Medicine, Faculty of Medicine, Lund University, 22184 Malmo, Sweden; 5USAID Population and Family Health Office, Amman 11183, Jordan; malkhateeb@USAID.gov; 6Infectious Disease Detection and Surveillance (IDDS), Rockville, MD 20894, USA; erik.reisdorf@icf.com

**Keywords:** health policy, molecular diagnostics, coronavirus, severe acute respiratory syndrome coronavirus 2, real-time polymerase chain reaction

## Abstract

Coronavirus disease 2019 (COVID-19) pandemic control measures rely on the accurate and timely diagnosis of infected individuals. Real-time polymerase chain reaction (qPCR) remains the gold-standard method for laboratory diagnosis of the disease. Delayed diagnosis due to challenges that face laboratories performing COVID-19 testing can hinder public health control measures. Such challenges may be related to shortages in staff, equipment or materials, improper inventory management, flawed workflow, or long turnaround time (TAT). The aim of the current study was to assess the overall COVID-19 molecular testing capacity in Jordan as of April 2021. In addition, the study’s objectives included the identification of potential defects that could comprise the utility of the COVID-19 molecular testing capacity in the country. All laboratories certified by the Ministry of Health (MoH) in Jordan to conduct molecular testing for SARS-CoV-2 were invited to participate in this study. Data were obtained from the participating laboratories (those which agreed to participate) by either telephone interviews or a self-reported written questionnaire with items assessing the key aspects of COVID-19 molecular testing. The full molecular testing capacity in each laboratory was self-reported considering 24 working hours. The total number of participating laboratories was 51 out of 77 (66.2%), with the majority being affiliated with MoH (*n* = 17) and private laboratories (*n* = 20). The total molecular COVID-19 testing capacity among the participating laboratories was estimated at 574,441 tests per week, while the actual highest number of tests performed over a single week was 310,047 (54.0%, reported in March 2021). Laboratories affiliated with the MoH were operating at a level closer to their maximum capacity (87.2% of their estimated full capacity for COVID-19 testing) compared to private hospital laboratories (41.3%, *p* = 0.004), private laboratories (20.8%, *p* < 0.001), and academic/research laboratories (14.7%, *p* < 0.001, ANOVA). The national average daily COVID-19 molecular testing was 349.2 tests per 100,000 people in April 2021. The average TAT over the first week of April 2021 for COVID-19 testing was 932 min among the participating laboratories, with the longest TAT among MoH laboratories (mean: 1959 min) compared to private laboratories (mean: 333 min, *p* < 0.001). Molecular COVID-19 testing potential in Jordan has not been fully utilized, particularly for private laboratories and those belonging to academic/research centers. Supply-chain challenges and shortages in staff were identified as potential obstacles hindering the exploitation of full molecular testing capacity for COVID-19 in the country.

## 1. Introduction

Amid the current coronavirus disease 2019 (COVID-19) pandemic, the World Health Organization (WHO) has advocated the importance of molecular testing to control the spread of the disease [1,2]. Nucleic acid amplification technology (NAAT), often implementing real-time polymerase chain reaction (qPCR), is considered the gold-standard method for laboratory diagnosis of COVID-19 [3,4,5]. Nevertheless, the potential limitations of this approach should be considered, including the dependence on specimen collection, handling and processing, relatively high cost, and lack of a standardized consensus on the limits of detection for various assays [6,7,8]. The achievement of timely and accurate COVID-19 testing results is invaluable for devising and assessing the strategies needed for proper control of the pandemic [1,9,10].

Molecular testing for the detection of severe acute respiratory syndrome coronavirus 2 (SARS-CoV-2) is mainly performed for two purposes: the diagnosis of COVID-19, which will help in case management; or for its potential epidemiologic value, which would help to identify infected cases with subsequent prevention of the forward transmission of the virus [10].

Previous research has suggested the potential benefits of random mass testing for SARS-CoV-2, which can be helpful to control the spread of COVID-19; however, this approach can be limited by the low testing capacity in some countries, in addition to the high cost of such an approach [11,12,13]. The extreme costs of PCR testing for SARS-CoV-2 in the United States were presented by Du et al. in a recent modeling study that investigated the cost effectiveness of expanding COVID-19 testing [14]. The usefulness of mass testing for SARS-CoV-2 has been demonstrated in countries with a high COVID-19 testing capacity (e.g., China, South Korea, and Taiwan), which is manifested in the rigorous control of virus spread [15,16,17]. In contrast, other studies have pointed to the minimal benefit of mass testing for SARS-CoV-2, though citing the possible value of stratified periodic testing [18,19,20]. Regardless of the testing approach, building capacity for COVID-19 molecular testing provides an opportunity to address the immediate needs of the COVID-19 response and help in better preparedness for future pandemics [6].

The evaluation of COVID-19 testing capacity at the country level can be an important initial step in strategic planning to control the current pandemic and possibly be better prepared for future epidemics [21]. This assessment can be helpful to identify potential limitations and obstacles in molecular testing (particularly for SARS-CoV-2), which could hinder public health control measures and negatively impact patient management [22]. Examples of these challenges that may compromise molecular testing capacity include: (1) shortages in well-trained and skilled staff; (2) supply-chain challenges (in the form of shortages in reagents, consumables, analyzers, and personal protective equipment (PPE)); (3) below-bar optimization of workflow; (4) improper inventory management; and (5) burnout “physical and psychological exhaustion” among laboratory workers [1,23,24]. Lack of strategizing and collaboration (especially in resource-limited settings) and weak biosafety measures can represent additional challenges hindering molecular testing capacity [25].

The evaluation of COVID-19 molecular testing capacity can form the baseline for efforts needed to scale-up the number of tests that can be conducted reliably. Such an approach can help to evaluate the potential obstacles that may result in critical delays in sample processing or reporting of test results. Consequently, such challenges may have a negative impact on the successful management and prevention of the disease [26].

The COVID-19 epidemic in Jordan went through several phases, with limited transmission to a cluster of cases through March until August 2020. The early response to the COVID-19 epidemic in Jordan was viewed initially as a promising approach [27]. This response entailed heavy mitigation measures, including curfews, wide lockdowns, and the prohibition of gatherings for more than 20 people. However, the inevitable spread of SARS-CoV-2 at the community level started in August/September and peaked in November 2020 as the first wave of COVID-19 in the country materialized.

Previous investigation of the molecular epidemiology of SARS-CoV-2 in Jordan revealed the continuous emergence/introduction and spread of various genetic lineages in the country, with at least 19 different genetic lineages [28]. The first wave of community spread was dominated by two Jordanian lineages, namely B.1.1.312 and B.1.36.10 [28]. Molecular analyses also revealed that the alpha variant (lineage B.1.1.7) was introduced into the country from November 2020 and became the dominant lineage from New Year 2021 onwards [28]. Despite the initial widespread circulation of misinformation about the pandemic’s origin, as well as COVID-19 vaccine hesitancy struggles in the country, the number of vaccinated individuals who received two doses (Pfizer-BioNTech, Oxford/AstraZeneca, or Sinopharm) reached 31.3% of the total population as of mid-September 2021, with continuous availability of free COVID-19 vaccination services provided by the MoH in Jordan [29,30,31].

The expansion of molecular testing capacity in Jordan was initiated by the Ministry of Health (MoH) as early as March 2020 through different committees that put forth strategic plans and policies for COVID-19 molecular testing based on WHO strategies [32].

The aim of the current study was to evaluate the molecular testing capacity for COVID-19 diagnosis in Jordan as of April 2021. In addition, the study’s objectives included the identification of potential bottlenecks that could hinder the timely exploitation of the current testing qPCR platforms in the country.

## 2. Materials and Methods

To evaluate the proportion of COVID-19 molecular testing performed by the laboratories that participated in this study in relation to the total number of COVID-19 qPCR tests in Jordan, we used “Our World in Data” to guide this aim. The retrieved data covered the period starting from 1 November 2020, until 21 April 2021 [33]. The latest estimate of the Jordan population was retrieved from the Worldometers website (https://www.worldometers.info/world-population/jordan-population/, acceesed on 23 April 2021).

Key information from the participating laboratories was obtained by the first author using field visits, telephone interviews, and e-mail communication. This was conducted between 14 March 2021, and 16 April 2021. Laboratories that were invited to participate were those certified by the MoH to conduct molecular testing for SARS-CoV-2, which comprised a total of 77 as of April 2021. Those laboratories belonged to five different categories: MoH laboratories (17 eligible), university/research/specialized center laboratories (4 eligible), private laboratories (37 eligible), private hospital laboratories (18 eligible), and the laboratory affiliated with the Royal Medical Services.

Data collected included the average number of daily specimens tested for COVID-19 using qPCR over the previous week, the total number of COVID-19 tests conducted over the previous week, the total number of laboratory personnel capable of performing COVID-19 testing, the number of nucleic acid purification equipment, the number of qPCR analyzers, working hours per laboratory, the self-reported maximum daily COVID-19 testing capacity considering the concurrent number of laboratory workers, nucleic acid purification instruments, and qPCR analyzers. Turnaround time (TAT) was defined as the time from sample receipt at the laboratory until reporting of results.

To assess the possible factors that limited the maximum utility of COVID-19 molecular testing capacity among the participating laboratories, multinomial logistic regression analysis was performed. The total number of laboratory staff capable of performing COVID-19 testing was divided into two categories as follows: Category 1, with 1–5 laboratory workers; and Category 2, with 6 or more laboratory workers. The total number of qPCR analyzers and nucleic acid purification equipment were divided into two categories as follows: Category 1, with 0–2 analyzers, and Category 2, with more than 2 analyzers. For working hours, the classification was Category 1, partial (less than 24 h); and Category 2, full time (24 h).

### Statistical Analysis

We used the chi-squared test (χ^2^), the two-tailed Mann–Whitney *U* (M-W) test, one-way analysis of variance (ANOVA) with Bonferroni correction for multiple comparisons, and multinomial regression analyses as appropriate. Statistical significance was considered for *p* < 0.050, and all analyses were conducted using IBM SPSS Statistics for Windows, v22.0.

## 3. Results

### 3.1. General Description of the Participating Laboratories

The total number of MoH-certified laboratories performing COVID-19 molecular testing that agreed to participate in the study was 51, divided as follows: private laboratories (*n* = 20), MoH laboratories (*n* = 17), private hospital laboratories (*n* = 10), and university/research/specialized center laboratories (*n* = 4). No data were available from the Royal Medical Services. By location, the majority of participating laboratories were located in the Central region (Amman, Balqa, Zarqa, and Madaba; *n* = 36, 70.6%), followed by the Northern region (Irbid, Ajloun, Jerash, and Mafraq; *n* = 9, 19.6%), and the Southern region (Karak, Ma’an, Tafilah, and Aqaba; *n* = 5, 9.8%, Table 1).

The only laboratories that were operating in the Southern regions belonged to the MoH, while private hospital laboratories were only operating in the Central region (Table 1). At the time of data collection (April 2021), 25 participating laboratories were involved in proficiency testing that runs through the MoH.

The total number of laboratory staff capable of performing COVID-19 testing varied, with an overall mean number per laboratory of 11 (median: 4, interquartile range (IQR): 3–8, range: 2–207). Stratified by sector, the results are shown in (Table 2).

The total number of qPCR analyzers varied as well, with an overall mean number per laboratory of 4 (median: 2, IQR: 1–4, range: 1–37). For the nucleic acid purification equipment, the mean number per laboratory was 3 (median: 1, IQR: 1–3, range: 0–27), and for the number of class II biosafety cabinets, the mean and median were 2 (IQR: 1–2), and the maximum number per lab was 12).

### 3.2. Assessment of the Total Number of Weekly COVID-19 qPCR Tests in Jordan during the Peaks of the First and Second Waves of COVID-19

The average number of daily tests performed in Jordan over the period (1 November 2020–22 April 2021) was 27,247 (median: 24,714). The average number of weekly tests over the same period was 191,477 (median: 163,510). The highest number of tests performed in a single day was 54,052 (on 11 March 2021), while the lowest number of daily tests was 9583 (on 15 November 2020). The highest number of weekly tests was reported in the week (8 March 2021–14 March 2021) and was 310,047 (Figure 1).

### 3.3. National Average Daily COVID-19 Tests Increased from 225 Tests per 100,000 People in November 2020 to 349 Tests per 100,000 in April 2021

Based on the latest estimate of the Jordan population (10,203,134 people), in the first week of April 2021 (5 to 11 April), the national average daily COVID-19 molecular tests was 35,628/10,203,134, resulting in 349.2 tests per 100,000 people, which was significantly higher than the average number of daily tests over the first week of November 2020 (2 to 9 November), reported at 22,962/10,203,134 (225.0 tests per 100,000 people; *p* = 0.011, M-W).

### 3.4. Daily and Weekly COVID-19 Testing Parameters among the Participating Laboratories in April 2021

The COVID-19 molecular testing parameters based on the first week of April (5 to 11 April) for the participating laboratories indicated that the total number of weekly tests was 225,346, considering the total number of tests 249,394 reported on the week ending on 11 April 2021. Thus, the participating laboratories in this study contributed to a total of 90.4% of all molecular COVID-19 tests conducted in Jordan by the first week of April.

MoH laboratories contributed to the majority of COVID-19 molecular tests (*n* = 178,312, 79.1%), followed by private laboratories (*n* = 35,677, 15.8%; Figure 2).

Based on the daily average of COVID-19 molecular tests performed on the first week of April 2021, the same pattern was seen with MoH laboratories contributing to 80.7% of all tests (*n* = 36,822), followed by private laboratories (*n* = 6759, 14.8%). About three-quarters of these tests were conducted in the Central region (*n* = 34,338, 75.3%), followed by the Northern region (*n* = 7616, 16.7%) and the Southern region (*n* = 3667, 8.0%).

### 3.5. Estimated Maximum COVID-19 Testing Capacity in Jordan

Based on the estimated data obtained from the participating laboratories, the maximum COVID-19 testing capacity in the country by the first week of April 2021 was 82,063. The actual tests performed on a daily basis over the same period were 45,621 (55.6%), resulting in 44.4% unexploited COVID-19 molecular testing capacity.

However, the deficit in exploitation of full testing capacity varied depending on the sector and location of the participating laboratories (Figure 3). MoH laboratories were operating at a level closer to their highest COVID-19 molecular testing capacity (36,822/42,244, 87.1%), compared to private hospital laboratories (1501/3630, 41.3%; *p* = 0.004), private laboratories (6759/32,520, 20.8%; *p* < 0.001), and finally university/research/specialized centers (539/3669, 14.7%; *p* < 0.001, ANOVA with Bonferroni correction for multiple comparisons).

### 3.6. Potential Factors Limiting the Utilization of Maximum COVID-19 Molecular Testing Capacity in Jordan

To assess the potential factors contributing to the deficit in the utilization of testing COVID-19 molecular testing capacity in relation to the current average daily COVID-19 tests, we divided the laboratories based on the deficit into two categories (Category 1: 0–50% deficit, and Category 2: 51–99% deficit).

Multinomial logistic regression analysis did not show any statistically significant differences in unexploited COVID-19 testing capacity between the laboratories with 0–50% deficit and those with 51–99% deficit, based on the number of laboratory workers capable of performing COVID-19 testing (*p* = 0.609), number of qPCR analyzers (*p* = 0.543), number of nucleic acid purification instruments (*p* = 0.287), and working hours (*p* = 0.851).

Self-reported limiting factors (with the ability to report multiple factors) for achieving maximum molecular testing capacity for COVID-19 included shortages in staff (100% among MoH laboratories and university/research/specialized centers laboratories, 60.0% among private hospital laboratories, and 55.0% among private laboratories). Shortages in equipment or testing kits were reported by 100% of university/research/specialized centers laboratories, 75.0% of private laboratories, 70.6% of MoH laboratories, and 50.0% of private hospital laboratories. A single laboratory reported that lack of PPEs was a limiting factor (private laboratory).

### 3.7. Factors Affecting the Average Daily COVID-19 Molecular Tests Performed in Jordan

The total number of laboratory workers divided into two categories showed that the mean number of tests performed by the laboratories having a staff of more than 5 was significantly higher than the laboratories having 1–5 staff capable of performing COVID-19 testing (mean: 2043 vs. 268; *p* < 0.001, M-W). Additionally, the number of qPCR analyzers was associated with a higher number of tests for the laboratories having more than 2 analyzers compared to those having 1–2 machines (average number of daily tests: 1714 vs. 408; *p* = 0.045, M-W). Moreover, the higher number of nucleic acid purification equipment was also significantly associated with a higher average number of daily COVID-19 tests among the laboratories having more than 2 instruments (2423 vs. 258; *p* < 0.001, M-W). Finally, the comparison of the average daily tests based on working hours did not show a statistically significant difference among the laboratories working 24 h compared to those with 16 h or less (643 vs. 964; *p* = 0.336, M-W).

### 3.8. Descriptive Statistics of TAT for COVID-19 Testing in Jordan

The average TAT among the participating laboratories over the first week of April 2021 for COVID-19 testing was 932 min (15.5 h, median: 420 min (7 h), IQR: 270–840 min (4.5–14 h), range: 60–7200 min (1–120 h)). Laboratories with an average TAT of 48 h or more were five, all of which were MoH laboratories.

The longest TAT was reported among MoH laboratories (mean: 1959 min) compared to private laboratories (mean: 333 min, *p* < 0.001) and private hospital laboratories (mean: 498 min, *p* = 0.013), while the difference was not statistically significant compared to university/research/specialized centers laboratories (mean: 649 min, *p* = 0.256, ANOVA with Bonferroni correction for multiple comparisons; Figure 4).

The total number of laboratory workers divided into two categories showed that the TAT among the laboratories with more than 5 workers capable of performing COVID-19 testing was significantly longer compared to the laboratories having 1–5 workers (mean: 1343 min vs. 708 min; *p* = 0.013, M-W). The number of qPCR analyzers was not associated with statistically significant differences between the laboratories having more than 2 analyzers compared to those having 1–2 machines (mean: 1147 min vs. 804 min; *p* = 0.107, M-W). The same applied for the number of nucleic acid purification equipment between the laboratories having more than 2 instruments compared to those with 0–2 instruments (mean: 1199 min vs. 820 min; *p* = 0.175, M-W). Finally, the comparison of the TAT based on working hours did not show a statistically significant difference among the laboratories working 24 h compared to those with 16 h or less (637 min vs. 1013 min; *p* = 0.810, M-W).

Multinomial logistic regression confirmed the lack of statistical significance for the comparisons in the paragraph above, with the location and health sector as covariates. This analysis was performed by dividing the TAT into two categories: the first for laboratories having a TAT less than 932 min, and the second for laboratories having a TAT of more than or equal to 932 min (the average TAT for the whole study sample).

## 4. Discussion

The major finding of this study was the demonstration of 44.4% unexploited COVID-19 molecular testing capacity in Jordan. However, large variability in testing capacity was found in different sectors, with the observation that MoH laboratories were operating at a level close to their maximum testing capacity (87%) compared to 21% in private laboratories. However, MoH laboratories had the longest TAT for COVID-19 qPCR tests, with an average of 33 h as opposed to the average of only six hours in private laboratories. In addition, the results of this study showed that molecular testing capacity in Jordan was mainly limited by shortages in staff, equipment, and testing kits based on self-reported data. However, the statistical assessment of such factors did not yield significant results, which points to the need for improved management of the available resources for molecular testing in the country.

The assessment of molecular testing capacity at the country level is important not only in the context of the COVID-19 pandemic [22]. The possibility of future pandemics requires preparedness and prompt vigorous response, including the ability to detect and trace infected cases at an early phase [34]. The utility of molecular testing appears crucial for rapid response to emerging infections [35]. Potential implications of the study findings include giving policymakers insightful clues on the importance of building up the capacity of molecular testing in resource-limited settings. In addition, the results of this study can help to highlight the existing difficulties that have precluded the utilization of full molecular testing capacity in epidemic situations, which in turn can help in a swift and more efficient response for future outbreaks. Moreover, our results hinted at the need for multisectoral involvement to optimize the response of laboratories to future epidemics at the national level.

In the Middle East, the previous encounter of another novel coronavirus (Middle East respiratory syndrome coronavirus (MERS-CoV)) highlights the possibility of outbreak occurrence resulting from emerging viral pathogens [36]. Thus, the current study was motivated by the need to evaluate COVID-19 molecular testing capacity in Jordan, which also reflects the ability to conduct molecular testing for other pathogens.

The value of this study is related to the identification of potential challenges that could hinder the utilization of full molecular testing capacity for viral infections, including SARS-CoV-2, in the country. Since nonpharmaceutical intervention to mitigate COVID-19 is still considered an important preventive measure, improving molecular testing capacity is highly needed, and the initial step in such an approach would provide an estimate of the current molecular testing capacity. This approach would be particularly helpful to prevent a resurgence in COVID-19 cases [37]. This estimation can also be used to enhance molecular testing capacity for large-scale surveillance [10]. Early on during the course of the current pandemic, the European Centre for Disease Prevention and Control advocated the importance of high levels of capacity for COVID-19 testing [38].

In Jordan and prior to the COVID-19 pandemic, molecular testing for viral infections was mainly conducted by a few specialized laboratories and research centers (about ten laboratories) and was limited by the high cost (MoH, personal communication). In March 2021, and during the peak of the second wave of COVID-19 in Jordan, the number of weekly COVID-19 tests performed reached more than 310,000, with more than 60 different laboratories conducting the test in all regions of the country. The rapid expansion in the number of sites conducting molecular testing for COVID-19 was shown by the findings of this study and was also reported in studies from Ethiopia, which demonstrated the expansion from zero to 65 laboratories as of October 2020, conducting up to 18,454 tests per day, which helped in improved case detection for COVID-19 [39,40].

Another study from Indonesia showed that achieving the WHO testing capacity target of 1 in 1000 inhabitants per week was reached in the country in July 2020 [26]. The challenges limiting the full utility of molecular testing capacity in Indonesia were listed by Hendarwan et al. and included workforce shortages, problems in logistics distribution, and complex administration, which were shown in another paper from Indonesia by Aisyah et al. [41]. In this study, the aforementioned factors were suggested by the participating laboratories as potential limitations hindering the exploitation of full COVID-19 molecular testing capacity. However, the results of the current study gave clues about the importance of optimal management of the allocated resources for COVID-19 testing as an important issue. This can be inferred by the lack of any discernible effect of such factors (working hours, number of staff, number of nucleic acid purification instruments, and qPCR analyzers) between the laboratories working at testing capacity ≤50% compared to those operating at testing capacity >50%. The importance of optimal allocation of limited resources, particularly for limited testing kits in outbreak settings, was demonstrated recently by Chatzimanolakis et al. in Switzerland [42].

In Ghana, Acheampong et al. showed the critical gaps preventing the development of laboratory capacity that were exposed during the COVID-19 pandemic [43]. These included weak biosafety and biosecurity measures and the limited budget for equipment and reagents. Thus, it appears that these shortcomings were common and prevalent, particularly in middle- and low-income settings.

In the current study, slightly more than half of the total molecular testing capacity for COVID-19 was utilized by the participating laboratories in Jordan. Self-reported limiting factors included shortages in equipment and reagents (supply-chain shortages). To expand laboratory testing capacity, a proper supply chain management system, quality assurance measures, management of laboratory equipment, and better biosafety measures and data management system are needed [6]. Early on during the course of the COVID-19 pandemic, shortages in skilled laboratory workers trained on molecular diagnostics techniques was an important limitation, together with problems in material and equipment procurement, shipping and handling, and stock management.

Thus, it appears that in various settings, the major bottlenecks hindering the timely molecular diagnosis of COVID-19 included shortages in reagents, the unreliability of some tests, and shortages in skilled staff, particularly in low-income settings [6,44,45].

Determining the defects and potential bottlenecks in COVID-19 molecular testing would be helpful to design a well-informed framework for viable and sustainable infrastructure that would help in rapid response to future pandemics, in addition to the support of clinical and surveillance values [46].

In this study, one important limitation of the utility of COVID-19 molecular testing was the relatively long TAT, particularly in MoH laboratories. Shortening the TAT would be highly valuable to inform contact tracing and for the isolation of infected patients. In this study, the median TAT was seven hours, which appears adequate for preventive action and clinical decisions to be taken. However, variability between the different sectors was noticed, with an average of 33 h in MoH laboratories, as opposed to only six hours in private laboratories. This issue should be considered with the aim of reducing the TAT for COVID-19 testing among MoH laboratories. However, “PCR is an easy thing to do badly”, with risk of contamination, unreliability, and necessity for rigorous quality control measures [47,48,49]. Thus, the aim of reducing the COVID-19 molecular testing TAT should be met with a meticulous approach in order to not compromise the reliability of results. Additionally, TAT estimation does not consider the time from sampling until receipt in the laboratory. Moreover, misinterpretation of data is another problem that should be considered with the inherent limitations of false-positive results [50]. Preanalytical problems and lack of consensus on the interpretation and determination of the threshold cycle values in qPCR further complicates the problem [51].

In the current study, only 11 laboratories reported adopting a 24 h working model. A study from India showed that moving to a 24 h working model can improve the daily testing capacity through maximum use of qPCR machines available with the multidisciplinary research units [52]. Another study from India by Mishra et al. showed the importance of collaboration and interaction among different institutes through the existing diagnostic, surveillance, and infection control networks [53]. In addition, Mishra et al. showed the importance of quality control measures in the building capacity of molecular diagnostics laboratories [53]. Our results indicated that proficiency testing provided through the MoH was only available in less than 50% of the participating laboratories. Different studies showed that the implementation of rigorous quality control measures is indispensable in any effort to scale-up molecular testing for COVID-19 [53,54,55].

In the current study, the various laboratories belonging to different sectors that participated in COVID-19 testing appeared advantageous. In Ecuador, a recent study showed the disadvantages of centralized qPCR testing, which resulted in critical delays in processing, which could hinder adequate clinical care and preventive efforts [56].

At the national level and in the context of infectious disease threats, strategic planning in the management of clinical laboratories at a multidimensional level appears vital, as shown in a recent review by Luo et al. [57]. This review demonstrated the importance of consolidation of regional clinical laboratories and the value of reasonable planning of laboratory resources [6,57].

Another study from the Americas advocated the importance of planning, collaboration, logistics, and training to enable a rapid response to future pandemics [58]. Collaboration at both the regional and country levels, involving public health laboratories was emphasized in the aforementioned study by Leite et al. to enhance laboratories’ proper response during the pandemic [58].

Even in high-income countries (e.g., U.S.), COVID-19 qPCR testing capacity did not meet the needs of the pandemic situation [59]. Thus, it is imperative to set up policies and operational plans and allocate adequate resources to allow for rapid scale-up of molecular testing capacity in case a future pandemic materializes, as suggested by El Hage et al. [60].

Other strategies to prepare for future pandemics were demonstrated in a recent preprint by Douthwaite et al., which suggested the potential value of incorporating new technologies, including heat inactivation of clinical samples upon receipt into the laboratory and with direct PCR, eliminating the need for RNA purification, which can increase the testing efficiency of molecular diagnostics [61,62].

Limitations of the study that should be considered in any future research with similar aims were as follows: the testing numbers are self-reported subjective estimates, and the accurate measurements might be slightly different, considering the subjective nature of reporting by various laboratories. In addition, the Royal Medical Services did not participate in this study, considering the difficulty in obtaining permission from their laboratories. However, we are inclined to believe that more than 90% of national COVID-19 molecular tests were included in this analysis based on the total number of tests reported in Jordan. In addition, the TAT reported in this study does not consider the time from sample collection until sample receipt in the laboratory conducting the test, which may impact the time from sample collection until reporting.

## 5. Conclusions

Despite reaching more than 300 daily tests per 100,000 people at the peak of the second wave of COVID-19 in Jordan, the molecular COVID-19 testing potential has not been fully utilized in the country, as estimated in this study. This unexploited testing capacity was more pronounced in private laboratories and those belonging to academic/research centers compared to MoH laboratories, a majority of which were operating close to their full molecular testing capacity for SARS-CoV-2. The self-reported defects identified in this study, which hindered the full utility of testing capacity for COVID-19, included supply-chain defects and shortages in staff. Optimal management of the allocated resources for COVID-19 molecular testing can be a helpful strategy to improve the testing capacity. Emphasis on the implementation of quality control measures such as proficiency testing should not be overlooked as well.

## Figures and Tables

**Figure 1 diagnostics-12-00909-f001:**
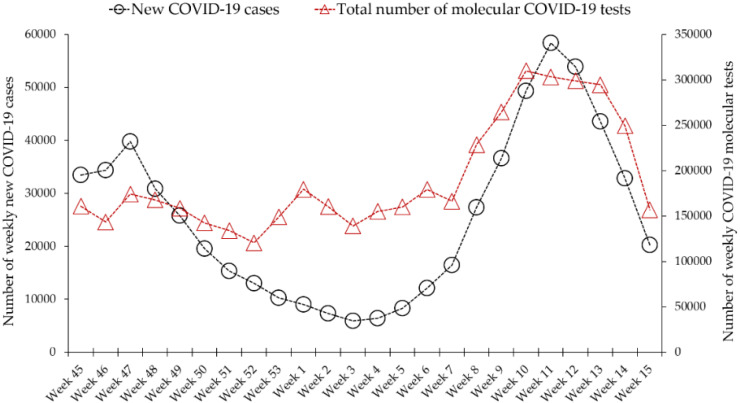
Total number of COVID-19 molecular tests performed in Jordan compared to weekly newly diagnosed cases. Data were retrieved from Our World in Data (https://ourworldindata.org/coronavirus/country/jordan, acceesed on 23 April 2021).

**Figure 2 diagnostics-12-00909-f002:**
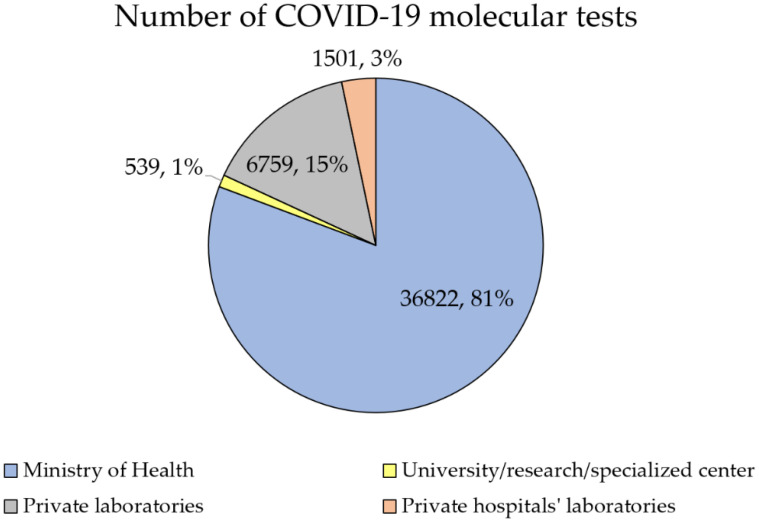
Total percentage of COVID-19 tests conducted by 11 April 2021, divided by health sector.

**Figure 3 diagnostics-12-00909-f003:**
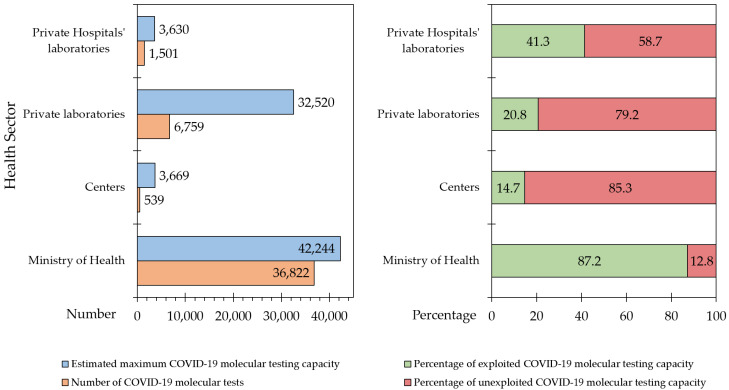
Estimated unexploited COVID-19 testing capacity per health sector in relation to the actual number of tests and full testing capacity. Centers: University/research/specialized center; COVID-19: Coronavirus disease 2019.

**Figure 4 diagnostics-12-00909-f004:**
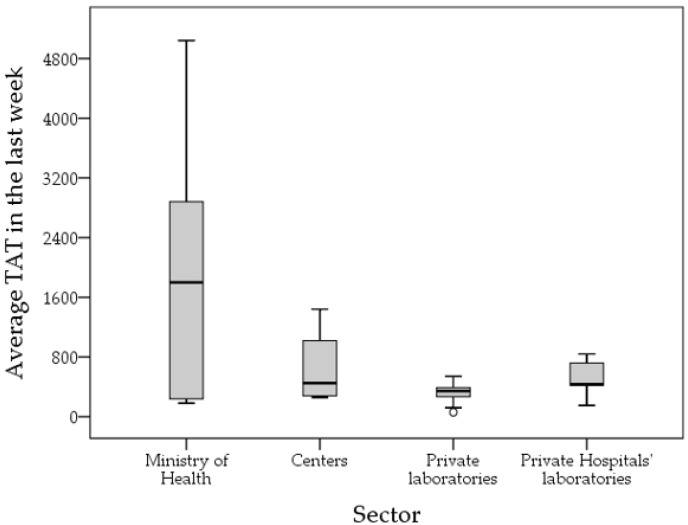
Average turnaround time (TAT) in minutes for the participating laboratories divided by health sector. Centers: university/research/specialized center. Last week: first week of April 2021.

**Table 1 diagnostics-12-00909-t001:** General features of the participating laboratories.

Region	Center	North	South
Health sector	*n* ^1^ (%)	*n* (%)	*n* (%)
Ministry of Health	7 (19.4)	5 (50.0)	5 (100.0)
University/research/specialized center	2 (5.6)	2 (20.0)	0
Private laboratories	17 (47.2)	3 (30.0)	0
Private hospital laboratories	10 (27.8)	0	0

^1^*n*: number.

**Table 2 diagnostics-12-00909-t002:** Number of laboratory staff capable of performing molecular COVID-19 testing stratified per health sector.

Health Sector	Descriptive Statistics	Staff Number
Ministry of Health	*n* ^1^	17
	Mean, median	8, 5
	Range	2–42
	IQR ^2^	4–8
University/research/specialized center	*n*	4
	Mean, median	8, 8
	Range	4–13
	IQR	5–12
Private laboratories	*n*	20
	Mean, median	17, 4
	Range	2–207
	IQR	3–12
Private Hospital laboratories	*n*	10
	Mean, median	5, 4
	Range	2–13
	IQR	3–5

^1^*n*: number; ^2^ IQR: interquartile range; COVID-19: Coronavirus disease 2019.

## Data Availability

The data presented in this study are available on a reasonable request from the first author (B.Q.).

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
