# Peer review of "Assessment of COVID-19 Molecular Testing Capacity in Jordan: A Cross-Sectional Study at the Country Level"

_diagnostics, 2022, doi:10.3390/diagnostics12040909_

Round 1
Reviewer 1 Report
Abstract:
- Generally, a study has one aim but could have more objectives.
- define "participating laboratories"
- 51 out of how many?
Define abbreviations in the keywords section.
Introduction:
- Appropritately present the limits of PCR in COVID-19 diagnosis.
- "the high cost of such an approach" list the extrems costs.
- It is not recommended to start a new paragraph with "Thus,"
- lines 99-102: you need to introduce here the information related with the availability of the vaccine.
- "The expansion of molecular" when?
- One study, one aim, more objectives.
Materials and Methods:
- "Data on new COVID-19 cases ..." when?
- "laboratories 117 that were invited to participate were those certified by the MoH to conduct molecular testing for SARS-CoV-2" how many?
- How many eligible laboratories for each category?
Results:
- The information presented in sections 3.2., 3.3., 3.4. is available online. You need to emphasize more why this information is needed in your article.
- Information in lines 220 - 226 belongs to the Methods section.
- "The total number of laboratory workers divided into two categories showed that the" this analysis in my personal view is obvious and no test is needed to show the significance. The same for equipments. These are only statistical exercises no science.
- "The average TAT over the first week" it is not clear if this reflect the experience of participating laboratories.
Discussion:
- It is useful to begin the discussion by briefly summarizing the main findings, and explore possible mechanisms or explanations for these findings.
- Emphasize the new and important aspects of your study with references to your own results. Put your findings in the context of the totality of the relevant evidence.
- State the limitations of your study, and explore the implications of your findings for future research and for clinical practice or policy.
- Discuss the influence or association of variables.
- Do not repeat in detail data or other information given in other parts of the manuscript, such as in the Introduction or the Results section.
Conclusions:
- State in this section only the conclusions that are supported by your results.
Author Response
Reviewer #1 Comments and Suggestions for Authors
- Abstract:
Generally, a study has one aim but could have more objectives.
Response: We are thankful for this comment and accordingly, we edited the abstract as follows: “The aim of the current study was to assess the overall COVID-19 molecular testing capacity in Jordan, as of April 2021. In addition, the study objectives included the identification of potential defects that could comprise the utility of COVID-19 molecular testing capacity in the country.” Lines 20-23.
- define "participating laboratories"
Response: We are thankful for this comment and accordingly we modified the abstract as follows: “All of the laboratories certified by the Ministry of Health (MoH) in Jordan to conduct molecular testing for SARS-CoV-2 were invited to participate in this study. The data were obtained from the participating laboratories (those which agreed to participate) by either telephone interviews or self-reported written questionnaire with items assessing the key aspects of COVID-19 molecular testing.” Lines 23-27.
- 51 out of how many?
Response: We are thankful for this important note by the reviewer, and accordingly we modified the abstract as follows: “The total number of participating laboratories was 51 out of 77 laboratories (66.2%)”. Line 29.
- Define abbreviations in the keywords section.
Response: Based on the reviewer’s comment we modified the Keywords as follows: “Health policy; molecular diagnostics; coronavirus; severe acute respiratory syndrome coronavirus 2; real-time polymerase chain reaction”. Lines 45-46.
- Introduction:
Appropritately present the limits of PCR in COVID-19 diagnosis.
"the high cost of such an approach" list the extrems costs.
Response: We would like to thank the reviewer for this important comment and accordingly we added the following paragraphs to the Introduction section: “Nevertheless, potential limitations of this approach should be considered and include the dependence on specimen collection, handling and processing, relatively high cost and the lack of a standardized consensus on the limits of detection for various assays [6-8].” Lines 53-56.
- Vandenberg, O.; Martiny, D.; Rochas, O.; van Belkum, A.; Kozlakidis, Z. Considerations for diagnostic COVID-19 tests. Nature reviews. Microbiology 2021, 19,(3): 171-183, doi:10.1038/s41579-020-00461-z.
- Arnaout, R.; Lee, R.A.; Lee, G.R.; Callahan, C.; Cheng, A.; Yen, C.F.; Smith, K.P.; Arora, R.; Kirby, J.E. The Limit of Detection Matters: The Case for Benchmarking Severe Acute Respiratory Syndrome Coronavirus 2 Testing. Clinical infectious diseases : an official publication of the Infectious Diseases Society of America 2021, 73,(9): e3042-e3046, doi:10.1093/cid/ciaa1382.
- Kanji, J.N.; Zelyas, N.; MacDonald, C.; Pabbaraju, K.; Khan, M.N.; Prasad, A.; Hu, J.; Diggle, M.; Berenger, B.M.; Tipples, G. False negative rate of COVID-19 PCR testing: a discordant testing analysis. Virology Journal 2021, 18,(1): 13, doi:10.1186/s12985-021-01489-0.
“The extreme costs of PCR testing for SARS-CoV-2 in the United States were presented by Du et al in a recent modelling study that investigated the cost-effectiveness of expanding COVID-19 testing [14].” Lines 67-69.
- Du, Z.; Pandey, A.; Bai, Y.; Fitzpatrick, M.C.; Chinazzi, M.; Pastore y Piontti, A.; Lachmann, M.; Vespignani, A.; Cowling, B.J.; Galvani, A.P., et al. Comparative cost-effectiveness of SARS-CoV-2 testing strategies in the USA: a modelling study. The Lancet Public Health 2021, 6,(3): e184-e191, doi:10.1016/S2468-2667(21)00002-5.
- It is not recommended to start a new paragraph with "Thus,"
Response: Based on the reviewer’s comment this word was deleted.
- lines 99-102: you need to introduce here the information related with the availability of the vaccine.
Response: Based on the reviewer’s comment, we added the following statement in the Introduction section: “with continuous availability of free COVID-19 vaccination services provided by MoH in Jordan”. Lines 112-113.
- "The expansion of molecular" when?
Response: Based on the reviewer’s note, we added the following statement in the Introduction section: “as early as March 2020,”. Line 115.
- One study, one aim, more objectives.
Response: Based on the reviewer’s suggestion, we modified the Introduction section as follows: “The aim of the current study was to evaluate molecular testing capacity for COVID-19 diagnosis in Jordan as of April 2021. In addition, the study objectives included the identification of potential bottlenecks that could hinder the timely exploitation of the current testing qPCR platforms in the country.” Lines 117-120
- Materials and Methods:
"Data on new COVID-19 cases ..." when?
Response: Based on the reviewer’s note, we added the following statement in the Methods section: “from 01 November 2020 till 21 April 2021”. Line 123.
- "laboratories 117 that were invited to participate were those certified by the MoH to conduct molecular testing for SARS-CoV-2" how many?
Response: We are thankful for this important note, and based on the reviewer’s comment, we added the following statement in the Methods section: “which comprised a total of 77 laboratories as of April 2021.” Lines 129-130.
- How many eligible laboratories for each category?
Response: We are thankful for this comment, and accordingly we modified the Methods section as follows: “Those laboratories belonged to five different categories: MoH laboratories (17 eligible laboratories), university/research/specialized center laboratories (4 eligible laboratories), private laboratories (37 eligible laboratories), private hospitals’ laboratories (18 eligible laboratories), and the laboratory affiliated to the Royal Medical Services.” Lines 129-133.
- Results:
The information presented in sections 3.2., 3.3., 3.4. is available online. You need to emphasize more why this information is needed in your article.
Response: We are thankful for this point and we would like to emphasize that the need for such data in our manuscript is as we intended to assess the expansion of COVID-19 molecular testing capacity during the peaks of different SARS-CoV-2 waves in the country. Thus, we prefer to keep the manuscript formatting regarding these sections as it is. However, based on the reviewer’s valuable suggestion, we added the following statement at the beginning of the sub-section 3.2 of the results section “To evaluate the proportion of COVID-19 molecular testing performed by the laboratories that participated in this study, in relation to the total number of COVID-19 qPCR tests in Jordan we used “Our World in Data” to guide this aim”. Lines 187-189.
- Information in lines 220 - 226 belongs to the Methods section.
Response: We are thankful for this comment, and accordingly we moved this paragraph to the Methods section. Lines 142-150.
- "The total number of laboratory workers divided into two categories showed that the" this analysis in my personal view is obvious and no test is needed to show the significance. The same for equipments. These are only statistical exercises no science.
Response: We are thankful for this comment; however, we disagree with the reviewer’s viewpoint. We believe that such analysis was necessary since the notion that the variability in each of the following factors (TAT, number of staff, number of analyzers, etc.) will be accompanied by subsequent increase in the exploitation of full molecular testing potential was not seen in the results of this study. Thus, we believe that this result has pointed to discrepancy in the self-reported factors that were cited by different laboratories for the inability to operate at maximum capacity for testing. Some antithetical results were found as well “The total number of laboratory workers divided into two categories showed that the TAT among the laboratories with more than 5 workers capable of performing COVID-19 testing was significantly longer compared to the laboratories having 1 ˗ 5 workers”. Thus, we believe that such analysis was necessary and we prefer to keep it if you agree with our viewpoint.
- "The average TAT over the first week" it is not clear if this reflect the experience of participating laboratories.
Response: We are thankful for this careful comment, and accordingly we modified this statement into “The average TAT among the participating laboratories over the first week of April 2021”.
- Discussion:
It is useful to begin the discussion by briefly summarizing the main findings, and explore possible mechanisms or explanations for these findings.
Response: We would like to thank the reviewer for this suggestion, and accordingly we added the following statement at the beginning of the Discussion: “The major finding of this study was the demonstration of 44.4% unexploited COVID-19 molecular testing capacity in Jordan. However, large variability in testing capacity was found in different sectors, with the observation that MoH laboratories were operating at a level close to their maximum testing capacity (87%) compared to 21% in the private laboratories. However, the MoH laboratories had the longest TAT for COVID-19 qPCR tests, with an average of 33 hours as opposed to the average of only six hours in the private laboratories. In addition, the results of this study showed that molecular testing capacity in Jordan was mainly limited by shortage in staff, equipment and testing kits based on self-reported data. However, the statistical assessment of such factors did not yield significant results, which points to the need for an improved management of the available resources for molecular testing in the country.” Lines 315-325.
- Emphasize the new and important aspects of your study with references to your own results. Put your findings in the context of the totality of the relevant evidence.
Response: We would like to thank the reviewer for this suggestion, and accordingly we modified the following parts of the Discussion sections: “In this study, the aforementioned factors were suggested by the participating laboratories as potential limitations hindering the exploitation of full COVID-19 molecular testing capacity. However, the results of the current study gave clues to the importance of optimal management of the allocated resources for COVID-19 testing as an important issue. This can be inferred by the lack of any discernible effect of such factors (working hours, number of staff, number of nucleic acid purification instruments and qPCR analyzers) between the laboratories working at testing capacity ≤50% compared to those operating at testing capacity >50%. The importance of optimal allocation of limited resources, particularly for limited testing kits in outbreak settings was demonstrated recently by Chatzimanolakis et al in Switzerland”. Lines 356-365.
“In this study, one important limitation of the utility of COVID-19 molecular testing was the relatively long TAT particularly in the MoH laboratories.” Lines 387-388.
“However, variability between the different sectors was noticed with an average of 33 hours in the MoH laboratories as opposed to only six hours in the private laboratories. This issue should be considered with the aim of reducing the TAT for COVID-19 testing among the MoH laboratories. However, “PCR is an easy thing to do badly”, with risk of contamination, unreliability and necessity for rigorous quality control measures. Thus, the aim of reducing COVID-19 molecular testing TAT should be done with a meticulous approach in order not to compromise the reliability of results.” Lines 387-398.
- State the limitations of your study, and explore the implications of your findings for future research and for clinical practice or policy. Discuss the influence or association of variables. Do not repeat in detail data or other information given in other parts of the manuscript, such as in the Introduction or the Results section.
Response: We would like to thank the reviewer for this valuable suggestion. However, we believe that the majority of the study limitations were mentioned in the Discussion section as follows: “Limitations of the study were as follows: the testing numbers are self-reported subjective estimates and the accurate measurements might be slightly different, considering the subjective nature of reporting by various laboratories. In addition, the Royal Medical Services did not participate in this study considering the difficulty in obtaining the permission from their laboratories. However, we are inclined to believe that more than 90% of national COVID-19 molecular tests were included in this analysis based on the total number of tests reported in Jordan. In addition, the TAT reported in this study does not consider the time from sample collection until sample receipt in the laboratory conducting the test, which may impact the time from sample collection till reporting.” Lines 438-446.
Regarding the implications of the results for future research and policy, and based on the reviewer’s comment this was mentioned in the Conclusions section as follows: “Optimal management of the allocated resources for COVID-19 molecular testing can be a helpful strategy to improve the testing capacity. Emphasis on the implementation of quality control measures including proficiency testing should not be overlooked as well.” Lines 456-459.
- Conclusions:
State in this section only the conclusions that are supported by your results.
Response: We would like to thank the reviewer for this comment. However, we believe that the conclusions section was a concise interpretation of the major findings of the study. Therefore, we prefer to keep it in the current format besides the suggested addition by the reviewer in the previous point.
Reviewer 2 Report
In the manuscript by Qaqish et al., the authors describe results from a cross-sectional study, evaluating the overall COVID-19 molecular testing capacity in Jordan and identifying potential defects that could comprise the utility of COVID-22 19 molecular testing capacity in the country with data obtained from the participating laboratories.
The topic could be interesting, I have some minor concerns.
Despite the authors discuss a lot about the timing of the study, in the paragraphs and in the tables but I would like to see the date of start and end of the study.
Furthermore, I would suggest the effort to give agility to the paper, removing and/or rephrasing some passages, nevertheless, I find a good paper with an interesting focus on the COVID-19 as an emergency management and in general a study on the capacity to meet the occurrence of emerging viral pathogens.
Author Response
Reviewer #2 Comments and Suggestions for Authors
- In the manuscript by Qaqish et al., the authors describe results from a cross-sectional study, evaluating the overall COVID-19 molecular testing capacity in Jordan and identifying potential defects that could comprise the utility of COVID-22 19 molecular testing capacity in the country with data obtained from the participating laboratories.
The topic could be interesting, I have some minor concerns.
Despite the authors discuss a lot about the timing of the study, in the paragraphs and in the tables but I would like to see the date of start and end of the study.
Response: We are deeply grateful for the positive critical appraisal of our manuscript. Based on the reviewer’s valuable comment we added the following statement in the Methods section: “This was conducted between 14-03-2021 till 16-04-2021.” Line 128.
- Furthermore, I would suggest the effort to give agility to the paper, removing and/or rephrasing some passages, nevertheless, I find a good paper with an interesting focus on the COVID-19 as an emergency management and in general a study on the capacity to meet the occurrence of emerging viral pathogens.
Response: We would like to thank the reviewer for the insightful comment; and we want to point to the following modifications in the Introduction and Discussion sections that hopefully will meet the reviewer’s request:
Lines 53-56; lines 67-69; lines 316-326; lines 357-366.
Thanks again for your positive appraisal of our manuscript.
Round 2
Reviewer 1 Report
Your manuscript looks better.
I have some minor suggestions that I believe will increase the readability of your manuscript:
- "Region." should be read as "Region" (Table 1)
- Table 2 - please ensure that you have all information on one page.
- "To evaluate the proportion of COVID-19 molecular testing performed by the laboratories that participated in this study, in relation to the total number of COVID-19 qPCR tests in Jordan we used “Our World in Data” to guide this aim [33]." this statement belongs to the Methods section. No references are accepted in the Results section for original studies.
- Please include the numbers and percentages on each column in Figure 3 to ensure an appropriate reading of the plots.
- Start the Discussion section with the paragraph in lines 317-327.
- Appropriately discuss the utility of the reported results for the scientific community and/or policymakers.
Author Response
Reviewer #1 Comments and Suggestions for Authors
Your manuscript looks better.
Response: We are deeply thankful for your valuable remarks that helped us to improve the quality of the manuscript.
I have some minor suggestions that I believe will increase the readability of your manuscript:
- "Region." should be read as "Region" (Table 1)
Response: Thanks a lot for this note. Correction was done as suggested.
- Table 2 - please ensure that you have all information on one page.
Response: Thanks a lot. Correction was done as suggested.
- "To evaluate the proportion of COVID-19 molecular testing performed by the laboratories that participated in this study, in relation to the total number of COVID-19 qPCR tests in Jordan we used “Our World in Data” to guide this aim [33]." this statement belongs to the Methods section. No references are accepted in the Results section for original studies.
Response: Thanks a lot for this important observation. The statement was moved into the Methods section as suggested. Lines 122-127.
- Please include the numbers and percentages on each column in Figure 3 to ensure an appropriate reading of the plots.
Response: We would like to thank the reviewer for this important suggestion and accordingly, we edited Figure 3. Please check page 8 of the revised manuscript (2nd revision).
- Start the Discussion section with the paragraph in lines 317-327.
Response: Thanks a lot for this note. The suggested revision was done. Please check page 10 of the revised manuscript (2nd revision). Lines 331-341.
- Appropriately discuss the utility of the reported results for the scientific community and/or policymakers.
Response: Thanks a lot for this insightful suggestion. Accordingly, we added the following paragraph in the Discussion section: “Potential implications of the study findings include giving the policymakers insightful clues to the importance of building up the capacity of molecular testing in resource-limited setting. In addition, the results of this study can help to highlight the existing difficulties that precluded the utilization of full molecular testing capacity in epidemic situation, which in turn can help in a swift and more efficient response for future outbreaks. Moreover, our results hinted to the need of multisectoral involvement to optimize the response of the laboratories to future epidemics at the national level.” Lines 346-353.